# Medically Significant Vector-Borne Viral Diseases in Iran

**DOI:** 10.3390/microorganisms11123006

**Published:** 2023-12-18

**Authors:** Sarah-Jo Paquette, Ayo Yila Simon, Ara XIII, Gary P. Kobinger, Nariman Shahhosseini

**Affiliations:** 1Department of Biological Sciences, University of Lethbridge, Lethbridge, AB T1K 3M4, Canada; sarahjo.paquette@alumni.uleth.ca; 2Department of Pathology and Laboratory Medicine, Faculty of Medicine, University of British Columbia, Vancouver, BC V6T 1Z4, Canada; simonyila28@gmail.com; 3Galveston National Laboratory, University of Texas Medical Branch, Galveston, TX 77555, USA; arxiii@utmb.edu (A.X.); gakobing@utmb.edu (G.P.K.)

**Keywords:** arboviruses, viral hemorrhagic fevers, climate change, epidemiology, mosquito, tick, sandfly, rodent, Iran

## Abstract

Vector-borne viral diseases (VBVDs) continue to pose a considerable public health risk to animals and humans globally. Vectors have integral roles in autochthonous circulation and dissemination of VBVDs worldwide. The interplay of agricultural activities, population expansion, urbanization, host/pathogen evolution, and climate change, all contribute to the continual flux in shaping the epidemiology of VBVDs. In recent decades, VBVDs, once endemic to particular countries, have expanded into new regions such as Iran and its neighbors, increasing the risk of outbreaks and other public health concerns. Both Iran and its neighboring countries are known to host a number of VBVDs that are endemic to these countries or newly circulating. The proximity of Iran to countries hosting regional diseases, along with increased global socioeconomic activities, e.g., international trade and travel, potentially increases the risk for introduction of new VBVDs into Iran. In this review, we examined the epidemiology of numerous VBVDs circulating in Iran, such as Chikungunya virus, Dengue virus, Sindbis virus, West Nile virus, Crimean–Congo hemorrhagic fever virus, Sandfly-borne phleboviruses, and Hantavirus, in relation to their vectors, specifically mosquitoes, ticks, sandflies, and rodents. In addition, we discussed the interplay of factors, e.g., urbanization and climate change on VBVD dissemination patterns and the consequent public health risks in Iran, highlighting the importance of a One Health approach to further surveil and to evolve mitigation strategies.

## 1. Introduction

Emerging and re-emerging vector-borne diseases are a major global concern, straining healthcare systems, hindering economic and social development both locally and on a global scale [1,2]. Vector-borne diseases (VBDs) represent more than 17% of all infectious diseases, which the World Health Organization (WHO) has estimated to cause over 700,000 deaths annually [3]. VBDs are increasing at an alarming rate with a disproportionate number of new illnesses being caused by viruses [1]. In fact, many viral hemorrhagic fevers (VHF) are vector-borne viral diseases (VBVDs) that transmit highly pathogenic agents from arthropods to humans and are responsible for deadly intercontinental outbreaks [4,5]. VBVDs are affected by a variety of peripheral factors, whether human, animal, or environmental (Figure 1), including climate change, urbanization, and travel.

VBVDs are spread by many vectors, including blood-sucking arthropods (e.g., mosquitoes, ticks, sandflies, and biting midges). Of the pathogens responsible for VBVDs, arboviruses cause countless animal and human infections worldwide. Recent data indicated that arthropod vectors such as mosquitoes or ticks, can carry more than one pathogen, leading to the co-transmission of diseases through one bite. This highlights the importance of VBVDs as a concern to public health [6]. Furthermore, while VBVDs are transmitted by a main vector, secondary vectors such as rodents often can transmit the same pathogens as insects, leading to multiple routes for infection [7].

**Figure 1 microorganisms-11-03006-f001:**
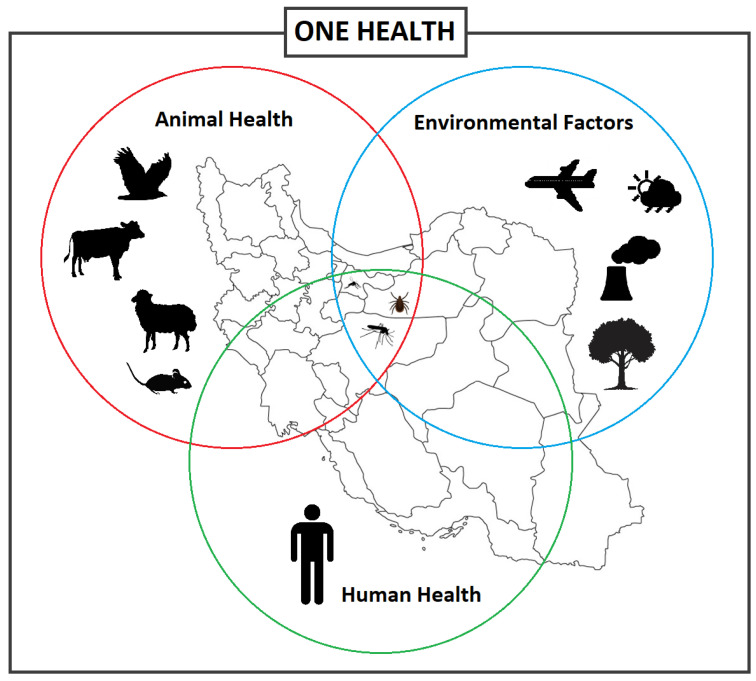
Different factors and interactions between the vector-borne diseases, vectors, and hosts including animals and humans, and environmental factors, informing the One Health concept.

A major factor influencing living and circulation patterns of pathogens and vectors are biospheric changes, particularly those involving climate systems. Ongoing climate change is affecting shifts in ecosystems and agricultural production systems [8], which in turn, alter VBVD transmission and incidence patterns [9]. Global warming has known correlations with increased development, survival, density, and expansion of ticks and mosquitoes. Furthermore, extreme weather events, such as floods, triggered by climate change, frequently have a sustaining effect on vector density. Countries in the Middle East are considered to be especially susceptible to the effects of climate change and the ensuing alterations of wet and dry climate bands [10]. In fact, within the next decades, temperatures in Iran are expected to rise 2.6 °C with a decline in precipitation, which can be anticipated to affect VBVD patterns.

The climatic changes effecting the Middle East are pronounced in Iran due to its vast geographical area, with various terrain and different climatic classification zones [2]. The country is home to various vectors and hosts and has the potential to be a major zone for VBVD outbreaks, as both Iran and many of its surrounding countries are endemic for VBVDs. Iran itself is endemic for several medically important VBVDs, such as Crimean–Congo hemorrhagic fever (CCHF) [11,12].

Several vectors of VBVDs including mosquitoes, ticks, sandflies, and rodents have been identified and are known to inhabit the natural fauna of Iran. Mosquitoes are known to be vectors for several medically significant public health VBVDs [13,14] including Chikungunya virus (CHIKV) [15], Dengue virus (DENV) [16], and West Nile virus (WNV) [17]. Iran is home to many different mosquito species, with 70 different species reported thus far [14,18]. Mosquitoes can survive and propagate under various climatic conditions [19]; therefore, increasing temperatures due to climate change is projected to have an extreme effect on distribution dynamics of infectious diseases [20]. Driving the increase in global distribution of disease is alterations to the expected ranges of mosquitoes by expanding populations to northern territories as temperatures rise or by broadening seasonal viability due to longer periods of warm weather.

In addition to mosquitoes, ticks are another an important vector of VBVDs identified in Iran [21]. Ticks are grouped into three families, specifically, Ixodidae (hard ticks), Argasidae (soft ticks), and Nuttalliellidae [22]. There are ~900 different tick species but only 46 have been reported in Iran [21,23]. Ticks can spread easily to new territories by attaching to animals, such as migratory birds traveling long distances to a new habitat [24]. Furthermore, in some cases, pathogens are maintained in a tick population through vertical transmission of the pathogen from females to their offspring (eggs) [25]. Ticks (along with mosquitoes) are key arthropods for the spread of disease to both humans and animals worldwide [22,26].

Sandflies are another important arthropod vector for VBVDs in Iran. While phlebotomine sandflies are well known for their association with leishmaniasis parasites, they are also the vector for Sandfly fever (SF), which is caused by *Phleboviruses*, referred to as sandfly fever viruses (SFV) [2,27]. Sandflies are small arthropods, rarely exceeding 3 mm, that live in habitats in close contact with animals or humans, whether it be rural or suburban [28].

Though VBVD definitions usually are confined to arthropods, rodents are an important vector of diseases playing a significant role in the transmission of a large number of viral diseases, both as a primary vector, such as for hantaviruses, or concomitant to arthropodal vectors [7,29]. Rodents are known to be reservoirs for over 60 zoonotic diseases and play a role in disease dissemination [30], either directly through the shedding of infected feces or urine [29] or indirectly by carrying disease from one location to another on their fur or paws [30,31]. Many diverse rodent populations, including Muridae and Cricetidae families, have been identified in Iran [7,32], which are known to maintain hantaviruses, the main viral disease spread by rodents in Iran [33]. However, other viral diseases, such as CCHF, that are typically spread by ticks, have been identified in rodents in Iran and can potentially also be spread by this alternate vector [7], which corroborates that one single vector can transmit multiple different diseases.

This review aims to provide an overview of four main vectors: mosquitoes, ticks, sandflies, and rodents, along with the main viral diseases they transmit in Iran. In addition, we discussed the interplay of factors, e.g., urbanization and climate change on VBVD dissemination patterns and the consequent public health risks in Iran, highlighting the importance of a One Health approach to further surveil and to evolve mitigation strategies.

## 2. Mosquito-Borne Viruses in Iran

### 2.1. Chikungunya Virus

Chikungunya virus (CHIKV), a mosquito borne virus (MBV), is a re-emerging *Alphavirus* that belongs to the Togaviridae family [34,35] and is deemed a worldwide threat that has caused epidemics in several countries [36]. Climate change, increased international trade, and increased travel are facilitating the local and global spread of the viruses. Viremic travelers can introduce the virus into regions where competent vectors are found, leading to potential autochthonous transmission, creating regions endemic for the disease, as demonstrated during the CHIKV outbreak in Italy [37].

CHIKV originated in Africa and was first detected in 1952 in Tanzania and isolated in 1953 [38,39]. The virus is grouped into 3 distinct lineages: West African, Asian, and East/Central/South/Africa [34]. CHIKV is a positive-sense single stranded RNA virus of approximately 11.8 kb with two open reading frames (ORFs). The initial ORF encodes four non-structural proteins and the other ORF encodes five structural proteins, which includes the capsid, as well as an envelope that is made of four proteins [34,36,37,40].

CHIKV infection is categorized as an acute febrile illness with an incubation period of four to seven days, similar to Dengue, Zika, and Malaria, leading to possible misdiagnosis, especially in the case of co-infections with other febrile illnesses [41]. Symptoms of Dengue are similar to CHIKV infection, making diagnosis challenging [42]. Patients with symptoms present with fever, severe arthralgia, skin rash, headaches, muscle aches, and joint pain [36,42]. CHIKV infection is seldom fatal in humans and in many cases resolves within 3 weeks; however, severe cases (10–15%) can lead to debilitating pain that can last weeks, if not years. Apart from being mosquito-borne, vertical transmission (mother to child) has been documented [42]. CHIKV is diagnosed through viral isolation using cell cultures, serological tests using antibodies, and RT-PCR tests using nucleic acids [34].

#### 2.1.1. Chikungunya Virus Situation in Iran

The first report of CHIKV circulation in the Middle East was recorded in Pakistan (1981), with recurring outbreaks in Pakistan, Saudi Arabia, and Yemen [43,44,45]. CHIKV is assumed to be present in Iran due to the presence of the virus in neighboring countries such as Pakistan [36,46] (Figure 2). During the 2017 outbreak in Pakistan, travel from Pakistan likely introduced the virus to Iran [36,46]. Several studies have been undertaken to detect the virus in Iran. Pouriayevali et al., 2019, examined serum from 159 patients with suspected CHIKV illness in the Iranian province of Sistan and Baluchestan, which borders Pakistan [36]. The study revealed that 25.1% of the serum samples were positive for CHIKV. A similar study by Tavakoli et al., 2020, examined serum from patients from 7 different provinces (Khuzestan, Fars, Kerman, Ilam, Hormozgan, Bushehr, Sistan and Baluchestan) with 16.07% of the samples testing positive for CHIKV [15]. Another study in 2018 examining blood samples collected from children at a children’s hospital in Tehran identified 2.2% positivity rate for CHIKV [47]. Similarly, serum samples collected in six provinces (Bushehr, Hormozgan, Sistan and Baluchestan, Khuzestan, Gilan and Mazandaran) from 2017 to 2018 revealed that 1.8% of the samples tested positive for CHIKV [48]. Altogether, these studies provide strong evidence of circulation of the virus in Iran.

#### 2.1.2. Chikungunya Virus Vectors in Iran

Reports of CHIKV outbreaks in the countries neighboring Iran, such as Pakistan, Qatar, Yemen, Iraq, Turkey, and Saudi Arabia suggest an existential threat toward the emergence of CHIKV in Iran. Viremic individuals are a major source of CHIKV dissemination, which can further extend locally through the spreading of infected mosquitoes. The main CHIKV vectors are *Aedes aegypti* and *Aedes albopictus*, the latter being confirmed in the south-eastern regions of Iran bordering Pakistan [49]. Mosquitoes maintain the ‘sylvatic cycle’ in monkeys and other vertebrates as common reservoirs for the virus. CHIKV is similar to other arboviruses, such as Dengue virus (DENV) and Zika virus (ZIKV), which are transmitted via the urban transmission cycle between humans and mosquitoes [36]. Recently, an Iranian study undertaken in the Mazandaran, North Khorasan, and Fars provinces aimed at detecting arboviruses in mosquitoes identified CHIKV (Asian genotype) in 3 different mosquito species [46]. Interestingly, the three species identified to carry CHIKV, *Anopheles maculipennis s.l.*, *Culiseta longiareolata*, and *Culex tritaeniorhynchus* are not the main vectors for CHIKV (Figure 2). Furthermore, it was the first report of infection of CHIKV in *An. maculipennis s.l.*, and *Cx. tritaeniorhynchus*. Identification of new potential vectors and the presence of the main vectors for CHIKV in Iran suggests an increased risk for CHIKV outbreak in Iran.

### 2.2. Dengue Virus

Dengue fever (DF) is an MBV that is considered a major global threat [50]. Its causative agent, DENV, is the principal source of VHF infections worldwide, causing Dengue hemorrhagic fever (DHF) [51]. DF incidence has grown significantly over the past 20 years, with the number of cases reported to the WHO dramatically increasing from ~500,000 cases in 2000 to 5.2 million cases in 2019, with the death toll quadrupling in the last four years, from 960 in 2015 to 4032 in 2019 [52,53]. As the result of biological, environmental, and socioeconomic factors such as climate change, urbanization, and international travel, DENV has expanded to areas previously free of the virus [54]. An estimated 3.9 billion people live in Dengue-endemic countries, including Afghanistan and Pakistan, which share an international border with Iran.

DENV belongs to the family of *Flaviviridae* and genus *Flavivirus* and is an 11-kb single-stranded positive-sense RNA virus [50,54,55]. The genome encodes a polyprotein that is composed of 3 structural and 7 non-structural proteins [50,54,55] and is classified into 4 serotypes (DENV1–4) [56]. Dengue has a 4–10 day incubation with a wide spectrum of clinical signs and symptoms, often with unpredictable clinical outcomes. All four virus serotypes can cause febrile illness [57]. While the infection is self-limiting in the majority of cases and patients fully recover from mild symptoms such as fever, headache, rashes, and muscle/joint pain [50], a small percentage of patients develop severe diseases, mostly characterized by severe abdominal pain, cardiac/pulmonary/hepatic problems, and plasma leakage with or without hemorrhagic syndrome (DF or DHF). The hemorrhagic form can be easily misdiagnosed as another hemorrhagic disease. DENV can be diagnosed using ELISA tests (detection of antibodies/antigens), virus isolation, or RT-PCR tests, and DENV antibodies can be present for 2–3 months after infection [50,57,58].

#### 2.2.1. Dengue Virus Situation in Iran

Neighboring countries sharing a border with Iran, such as Afghanistan and Pakistan, are endemic for DENV, potentiating the spread into Iran [59]. DENV became a public health concern in Iran in 2008, after the first case of Dengue fever was reported in a patient who had previously travelled to Malaysia [60] (Figure 2). Similarly, two Iranians with confirmed DENV had a history of travel to Malaysia, during years of comparably high-infection rates (2009 and 2011, respectively) [61]. Another imported case of DENV into Iran was reported in 2015, when a patient with recent history of travel to India was admitted to a hospital with signs and symptoms of the disease and subsequently tested positive for DENV 2, genotype 4, similar to the strain isolated from patients in India [16]. These studies provide evidence on how international travel can have an impact on disease spread from endemic countries.

In a retrograde investigation in Iran, the presence of specific antibodies against DENV was observed in 5% of a previously studied population. Among the seropositive cases, 53% of the population had a travel history to Dengue endemic countries including Malaysia, India, and Thailand; conversely, 46% of the population did not [62]. However, those individuals with no travel history reside in Iranian provinces that share a border with either Pakistan (Sistan and Baluchestan province) or Iraq (Kurdistan province), suggesting the presence of DENV in Iran was due to proximity to endemic countries. Another study searching for the incidence of DENV among the healthy population in Chabahar city in south-eastern Iran showed ~6% of the studied population were seropositive for DENV [63]. A similar study on sera from patients presenting with rash and fever collected during 2016 and 2017 showed that 6.27% of the studied population were DENV seropositive [15]. Lastly, a study by Heydari et al., 2018, aimed to determine if serum samples of patients presenting with symptoms of DENV but negative for CCHF from 2013 to 2015 in the province of Sistan and Baluchestan, which borders Pakistan, were positive for DENV [58]. The study determined that 13 of the 60 patients were exposed to DENV, providing support that DENV is circulating in Southeastern Iran. Altogether, these results provide strong evidence that DENV is circulating in Iran, especially in provinces that border countries endemic for DENV.

#### 2.2.2. Dengue Virus Vectors in Iran

*Ae. aegypti* and *Ae. albopictus* are the two main vectors of DENV, which are endemic to tropical and subtropical climates. DENVs circulate between humans and vector mosquitoes with no intermediate host. Thus, the spatial distribution of the vectors highly affects the epidemiology of the disease [50,59,64]. The presence of *Ae. aegypti* was recently confirmed both morphologically and molecularly in southern Iran [65], and *Ae. albopictus* has been isolated both in 2009 and 2013 in Iran [49]. The *Ae. albopictus* mosquito is considered a conspicuously menacing and versatile species that inhabits both temperate and tropical climate regions. Another species, *Aedes unilineatus*, was identified also in the south-east of Iran (2012–2014) and, interestingly, has been previously reported to be the DENV vector in Karachi, Pakistan [66]. The presence of these mosquito vectors in the regions of Iran where previous seropositive human cases for DENV with no travel history was reported, indicates the risk of DENV outbreaks exists with the potential for autochthonous transmission to humans in this area.

### 2.3. Sindbis Virus

Sindbis fever is an illness transmitted by mosquitoes infected with the Sindbis virus (SINV) [67]. First isolated from mosquitoes in 1952 in Egypt, SINV belongs to the family *Togaviridae* and genus *Alphavirus* [34,67]. The virus is enveloped and is made of positive-sense, single-stranded RNA genome with a genome size of 11.7 kb that encodes four non-structural and five structural proteins [68]. SINV can be grouped into six different genotypes. SINV- I, which is associated with human outbreaks, has been isolated from the Middle East, among other countries.

The virus is widely distributed in Africa, Asia, Eurasia, and Oceania with sporadic outbreaks in Australia, China, and South Africa [69]. Genetic studies have showed that SINV strains isolated in Africa, Europe, and the Middle East are geographically distinct serotypes [70]. Despite SINV being identified across various continents, clinical infections are mainly reported in Europe and Africa [71]. Signs and symptoms of SINV include fever, rash, arthritis, and myalgia [71], which in most cases resolves within a few weeks [72]. However, in some cases, the arthritis and myalgia can persist for years. The chronic disease syndrome following SINV infection shares many features with autoimmune diseases, which is thought to be immune mediated [72]. SINV is typically identified using specific antibodies in human serum samples, as virus isolation has seldom been used successfully to identify SINV [71,73]. Antibodies can persist for several months post infection [71].

#### 2.3.1. Sindbis Virus Situation in Iran

SINV is a zoonotic disease transmitted from birds to humans via mosquitoes, which may spread to non-endemic countries by migratory birds [71,74]. A study undertaken by Hanafi-Bojd et al., 2021, identified SINV in both *Culex pipiens* complex and *Culex theileri* in the West Azerbaijan province of Iran [75], with both mosquito species distributed across the country (Figure 2). Furthermore, these mosquito habitats are suitable also for the migratory birds participating in the zoonotic cycle of SINV. Altogether, these results strongly suggest the virus has a high potential to spread to humans in the area, though, thus far, only two human cases of SINV have been reported in Iran.

#### 2.3.2. Sindbis Virus Vectors in Iran

Interestingly, SINV mosquito vectors are different than that of CHIKV and DENV, as SINV is mainly transmitted by *Culex* mosquitoes [68,74]. In 1952, SINV was first isolated from *Cx. pipiens* and *Cx. univittatus* [76]. Over the years it has been determined that the *Aedes*, *Culiseta*, and *Culex* mosquito species are all viral vectors that help maintain the transmission cycle between wild birds and humans. In Iran, the mosquitoes that transmit SINV are readily identified throughout the country, which is a cause for concern, making monitoring of the disease and its vectors of crucial importance [75,77,78]. Furthermore, studies show that SINV is transmitted from *Culex* females to their offspring, facilitating disease maintenance [79].

### 2.4. West Nile Virus

West Nile virus (WNV) is a zoonotic *flavivirus* that is part of the Japanese encephalitis virus complex [80,81]. It is both a human and animal pathogen, notably infecting birds and horses. WNV has global public health implications to both humans and animals, making it a One Health priority concern [81,82]. WNV is a single-stranded RNA virus of approximately 11 kb and contains a single ORF that encodes three structural and seven non-structural proteins [80,81]. WNV is grouped in eight distinct lineages. Lineage 1 consists of two sub-lineages: clade 1A is formed by widespread strains and clade 1B by Australian WNV Kunjin strains. Lineage 2 is distributed in Sub-Saharan Africa and Europe. Lineage 3 circulates in the Czech Republic. Lineage 4 was detected from the Caucasus. Lineage 5 contains a WNV isolate from India [83]. Lineage 6 contains a Malaysian Kunjin virus [84]. The African Koutango virus is closely related to WNV and considered lineage 7. WNV lineage 8 has been reported from Spain [85,86]. Finally, lineages 1, 2, and 5 are known to cause human infection [87].

WNV was first detected in 1937 in the blood of a febrile woman in the West Nile region of north-western Uganda [88]. Over the past few years, several outbreaks have occurred in Europe [89], including a large outbreak with over 2000 cases in 2018 [90]. The incubation period for WNV infection in humans ranges from 3–15 days [91,92]. Most infected humans (75–80%) have very mild signs and symptoms of the disease and are either asymptomatic or have mild fever. Approximately, 20% of infected humans have fever and flu-like symptoms, and less than 1% of infected humans have neuroinvasive complications [81,87]. Neuroinvasive disease is characterized by meningitis, encephalitis, and movement disorders that can have lifelong implications [81]. WNV is typically detected using diagnostic approaches including virus isolation, RT-PCR, serological assays using antibodies, and pathological tests examining tissues microscopically.

#### 2.4.1. West Nile Virus Situation in Iran

Human cases of WNV occur globally, including in Iran [93]. The first human cases of WNV in Iran were detected in 1976 in the central and southwestern regions of Iran [94] (Figure 2). Since then, various diagnostic studies have been undertaken to identify WNV in Iran. Two serological surveys occurred in the 2000s, one that determined 5% of healthy blood donors in Tehran city were serologically positive for WNV [95]. The second identified that 15% of migratory/resident wild birds were seropositive for WNV in north-western Iran [96]. WNV was identified as the causative agent for encephalitic signs in the cerebrospinal fluid (CSF) of Iranian patients in Isfahan city in 2008 and 2009 [97]. Subsequently, this strain was determined to be genetically related to the lineage 2 Central African Republic WNV strain [98].

Studies published in the last 10 years are still identifying substantial WNV infections in Iran. Serological studies undertaken from 2010 to 2012 have demonstrated that WNV was observed in 1.3% of human samples and 2.8% of equine samples from five provinces in North and Central Iran [99]. Additionally, 11% of the investigated population residing in Mashhad city in north-eastern Iran were positive from previous exposure to WNV [93]. A study examining blood donors in the Iranian province of Sistan and Baluchestan identified WNV in 8.24% of blood donors, revealing the circulation of the disease in the human population in the province [100]. Another study by Ziyaeyan et al., 2018, in the Hormozgan province that examined human serum samples and mosquitoes for WNV revealed that 20.6% of the human samples were positive for WNV [101]. The virus was also identified in *Cx. pipiens* pools in the region, providing evidence of virus circulation in the province. Lastly, a study examining serum samples from birds and horses in 4 provinces: Kordestan, Mazandaran, Golestan, and North Khorasan identified WNV in ~14% of birds and ~17% of horses, indicating the presence of the virus in North Iran [17] (Figure 2). Altogether, these results provide evidence for the continuous presence and circulation of WNV in Iran, indicating a need for strong monitoring and protocols to minimize the potential of a large-scale outbreak in the country.

#### 2.4.2. West Nile Virus Vector in Iran

The primary mosquito vectors for WNV are the *Culex* mosquitoes. A large-scale study was undertaken to examine mosquitoes for the presence of arboviruses (including WNV). The study reported WNV lineage-2 in *Cx. pipiens pipiens* form *pipiens* (Cpp) in the province of Gilan [102,103] (Figure 2). Of note, the presence of WNV lineage 2 in Cpp and the major abundancy of this species in parts of Iran [104,105] lends support to this vector being responsible for both enzootic and epizootic transmission of WNV [80]. Another study identified WNV in an *Aedes caspius* mosquito species [96], and subsequent studies identified these as lineage 1 WNV strains [106], suggesting co-circulation of WNV lineages in Iran. In addition, Shahhosseini et al. identified WNV in *Cx. theileri*, one of the most common mosquito species in Iran, which suggests this may be a new vector for WNV within the country [107]. Furthermore, a recent study in southern Iran provided evidence that WNV can be transmitted vertically between female *Culex* mosquitoes and their offspring, which may explain how the virus survives the winter months. This consideration expands concerns beyond an enzootic seasonal consideration, making WNV an even greater public health and One Health concern in Iran [108].

### 2.5. Other Related Mosquito-Borne Viruses in Iran

#### 2.5.1. Rift Valley Fever Virus Situation in Iran

Rift Valley Fever virus (RVFV) is a mosquito-borne *Phlebovirus* that mainly affects ruminants but can also infect humans [109,110]. The primary vectors for RVFV are the *Aedes* mosquitoes but *Culex*, *Anopheles*, and *Mansonia* are secondary vectors [111], all of which have been identified in Iran [14,18]. RVFV disease in humans vary from a mild febrile illness that resolves in a few days to acute forms causing severe signs and symptoms such as blindness, encephalitis, and death [110,111].

RVFV is widespread in Africa, with spillover to the Comoros Archipelago (including Mayotte), Madagascar, Saudi Arabia, and Yemen [112]. Following the outbreak of RVF in Saudi Arabia in 2000, a surveillance of both animal and human populations was conducted in neighboring Iran from 2001 to 2011. While human cases of RVFV have not been identified in Iran [113], the virus antibody (IgG) has been identified in cattle and sheep in the province of Kurdistan [109] (Figure 2). Recently, another investigation to study associated risk factors in aborted sheep in Kurdistan province showed that 1.65% of fetal abortions in sheep may be linked to seropositivity to RVFV [109]. Furthermore, sandflies and tabanids, which are present in Iran, have been shown to participate in virus transmission, increasing the potential vectors for RVFV [114,115]. Thus far, although direct evidence of RVFV circulation in Iran is lacking, continued enhanced surveillance remains a One Health concern toward revealing the real status of RVFV in Iran and preventing a potential spillover of RFVF to humans there.

#### 2.5.2. Zika Virus Situation in Iran

Zika virus (ZIKV) is a mosquito-borne *Flavivirus* that gained notoriety in 2015 when it spread from South America to North America [116]. ZIKV infection is not fatal but can cause serious complications, especially for children born from infected mothers [117]. Presently, ZIKV has not been identified in Iran; still, surveillance has been enhanced, especially due to a potential for travel-related cases of the disease [116,117,118]. A study in 2018 examining both human serum and mosquitoes in southern Iran did not find ZIKV, and the main vectors, *Ae. aegypti* and *Ae. albopictus*, were not identified in the mosquito samples [101]. Recently, however, a main ZIKV vector, *Ae. aegypti* was identified in southern Iran [65]. While ZIKV has not been identified, the presence of the vector is a cause for concern and surveillance should be continued to avoid a potential future outbreak. More studies examining ZIKV vectors need to be undertaken to elucidate the potential of disease spread in Iran, especially considering the presence of a main vector for the disease.

**Figure 2 microorganisms-11-03006-f002:**
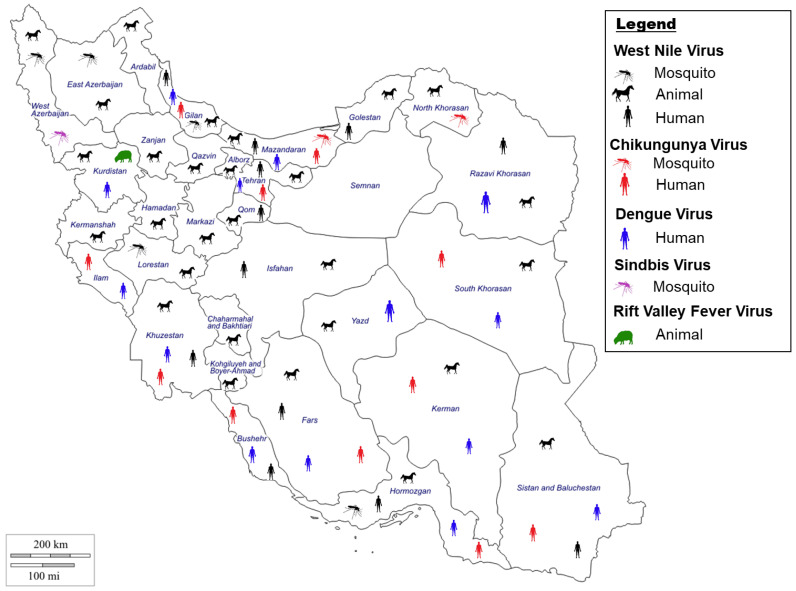
Evidence of West Nile virus [48,96,99,107,119,120,121,122], Chikungunya virus [15,46,47,48], Dengue virus [15,48,58,60,62], Sindbis virus [75] and Rift valley fever virus [109] in Iran in either humans, animals, or mosquitoes by indirect and direct evidence.

## 3. Tick-Borne Viruses in Iran

### 3.1. Crimean–Congo Hemorrhagic Fever Virus

Crimean–Congo Hemorrhagic Fever virus (CCHFV) is a zoonotic virus and one of the most important tick-borne virus (TBV) pathogens [23,123]. CCHFV was first discovered in 1944 in Crimea after soldiers became ill with a hemorrhagic disease [24,124]. Twenty years later, a virus with similar clinical signs was reported in the Congo, hence the multi-geographic name. Subsequently, CCHFV has been reported in other countries in Africa, the Middle East, the Balkans, and Asia [11,125]. Globally, 3 billion people are estimated to be at risk of infection, with 10,000–15,000 infections occurring annually and ~500 cases having fatal outcomes [126].

CCFHV belongs to the family *Nairoviridae* of the order *Bunyavirales.* The virus has a single-stranded negative sense RNA genome that consist of three segments: small (S), medium (M) and large (L), which encode the viral nucleocapsid, the glycoprotein precursor that is cleaved into two envelope glycoproteins (GN and GC), and the RNA polymerase. Genetic evidence reveals a recombination and re-assortment of genes through the evolution of CCFHV [127,128]. The virus can be grouped into seven distinct genotypes: Africa-1, Africa-2, Africa-3, Asia-1, Asia-2, Euro-1, and Euro-2 [126,129].

The worldwide distribution of CCHFV makes it the most widespread TBV that infects humans [9]. The spread of the disease is correlated with the distribution of its main vector *Hyalomma* ticks and their habitats, particularly forest land interspaced with shrub cover [11,124]. Although tick bites are the main route of transmission to humans, human infections have resulted from direct contact with the tissue or bodily fluids, such as blood, of infected animals during slaughtering or in the healthcare setting from infected patients [126,130]. Infected animals usually are asymptomatic but serve as a reservoir for the virus.

CCHFV infections can develop into severe hemorrhagic fever in humans, with a death rate of ~50% on average [24,131]. Typically, mild signs and symptoms of CCHF are headache, fever, nausea, myalgia, vomiting, and joint pain [124,131]. However, in severe cases, CCHF disease results in a sudden onset of fever, progressing to hemorrhages and associated bruising, with possible death within days of onset. CCHFV is diagnosed using serological assays to detect antibodies, RT-PCR for viral genome detection, and virus isolation [11].

#### 3.1.1. Crimean–Congo Hemorrhagic Fever Virus Situation in Iran

CCHFV was first discovered in Iran in the 1970s in serum from animals and humans but was not diagnosed in a patient until 1999 [132,133,134], and since then has become endemic to the country [135] (Figure 3). CCHFV has been detected in humans and livestock in most provinces in Iran [136,137,138]. Iran is surrounded by countries endemic to CCHF, with Sistan and Baluchestan province, which shares borders with Pakistan and Afghanistan, the main center of CCHFV infection in Iran [139,140]. Furthermore, Khorasan and Isfahan provinces also showed a high prevalence for CCHF. The higher prevalence potentially can be correlated to the region’s long and warm summers, which create a suitable environment for tick populations to propagate and survive for longer periods of time [132,141].

Several studies have been undertaken to identify CCHFV in Iran. A serological study conducted from 2000 to 2004 in various Iranian provinces determined that 36.1% of suspected cases were positive for CCHFV antibodies [137]. Since then, Chinikar et al. have undertaken numerous studies aimed at identifying CCHF in Iran in both humans and livestock; various reports are summarized below [142]. A 2003–2004 study in Sistan and Baluchestan identified a 6.3% seropositivity rate among human volunteers. Livestock surveys in Isfahan and Khorasan in the early 2000s identified a 56% seropositivity rate in animals in Isfahan, and in Khorasan a 77.5% and 46% seropositivity rate for CCHFV in sheep and goats, respectively. More recently, a study by Saghafipour et al., 2019, in north Khorasan, determined that CCHFV was present in both ticks and blood samples collected from domestic animals. The results of this study coupled with the previous study in Khorasan strongly suggest CCHFV is endemic to the province [143].

Aside from serological studies, several CCHF outbreaks have been recorded in Iran, mostly in the south-east and north-east of Iran [139,142]. A recent outbreak occurred in 2019, in Ardabil province (north-west Iran), when 51 blood samples were tested and 19.6% of patients were positive for CCHFV [141]. This was the first outbreak recorded in northwest Iran, which demonstrates disease spread, as typically cases are reported in south and south-east Iran. In addition, CCHF nosocomial cases were reported in several provinces across Iran [144,145,146].

Typically, the majority of CCHF confirmed cases in Iran are butchers, farmers, slaughterhouse workers, and housewives, while the death ratio is highest amongst housewives followed by farmers, slaughterhouse workers, and butchers [147,148]. Interestingly, CCHF infections rates are approximately three times higher in males compared to females, but females have a higher death rate. The higher infection rate in males may be due to their predominance in high-risk careers; in fact, nearly all slaughterers and butchers in Iran are male [149]. The common age range of CCHF patients is 20–40 years old, with a higher occurrence of CCHF patients in warmer seasons (June and July). Genetic characterization of CCHFV in Iran showed simultaneous circulation of four different CCHFV genotypes circulating in Iran, including Asia-1, Asia-2, Europe-1, and Europe-2 [150,151,152,153,154]. Altogether, these studies underscore the importance of surveillance to help mitigate outbreaks of CCHF, and further studies are needed to elucidate more areas of monitoring or spread of the disease, especially in light of agricultural activities, climate change, and population expansion, which may alter tick habitats.

#### 3.1.2. Crimean–Congo Hemorrhagic Fever Virus Vectors in Iran

Ticks, predominantly the *Hyalomma* genus of hard ticks, are the vector and reservoir for CCHFV and once infected maintain the disease for their lifetime [9,11]. CCHFV has been identified in other ticks, namely from the genera, *Rhipicephalus*, *Haemaphysalis*, *Amblyomma*, *Ixodes*, and *Dermacentor* [24]. Interestingly, both hard and soft ticks are found to harbor CCHFV in Iran [155,156]; in fact, CCHFV was first detected in Iran from the *Ornithodoros lahorensis* soft tick [134] (Figure 3). Based on several surveillance studies in Iran, CCFHV has been isolated from numerous different tick species, and the most common tick species carrying CCHFV across Iran includes *Hyalomma marginatum*, *Hyalomma anatolicum*, *Hyalomma asiaticum*, *Hyalomma dromedarii*, *Rhipicephalus sanguineus*, and *Dermacentor marginatus* [157,158].

The distribution of CCHFV-positive ticks varies by province and species. In northwest Iran (Ardabil province), tick species such as *Rhipicephalus bursa*, *Hyalomma genus*, *Or. lahorensis* were positive for CCHFV [159]. In the Hamadan province, several species collected on livestock such as *Hyalomma* genus, *Rh. sanguineus*, and *Argas reflexus* were positive for CCHFV [160]. In western Iran (Kurdistan province), CCHFV positive ticks of the *Hyalomma* genus were collected on sheep, goats, and cattle [161]. Other studies have shown CCFHV-positive tick species belonging to the genus *Hyalomma* and *Rhipicephalus* on cattle, goat, and sheep in Kermanshah and Sistan and Baluchestan provinces (Southeastern Iran) [140,162], and *Hy. dromedarii* on camel in Khorasan province (north-eastern Iran) [163]. CCHFV-positive ticks are constantly being detected among livestock populations in new geographical areas in Iran, supporting the endemicity of the virus in Iran [158,164]. Furthermore, a spatial-distribution study demonstrated that CCHFV-positive ticks from 6 genera and 16 species of hard and soft ticks were identified in 18 of 31 Iranian provinces [23].

### 3.2. Other Related Tick Borne Viruses in Iran

#### 3.2.1. Zahedan Rhabdovirus Situation in Iran

Climate change affects all ecosystems and can have an effect on VBVDs patterns as well, especially by expanding the range of vectors such as ticks and mosquitoes [9] and potentially creating suitable environments for the emergence of new diseases. Recently, a new TBV was identified in Iran and coined, Zahedan Rhabdovirus (ZARV) [165] (Figure 3). ZARV is a 11 kb rhabdovirus with a negative single-stranded RNA genome. The virus was isolated from a *Hy. anatolicum anatolicum* tick in Iran, and genetic comparison suggests ZARV, Moussa virus (isolated from *Cx. decens* in Côte d’Ivoire), and Long Island tick rhabdovirus (isolated from *Amblyomma americanum* ticks in the USA) form a monophyletic group of arthropod-borne rhabdoviruses. An animal reservoir has not been identified for ZARV in Iran, and more research is needed to determine reservoirs and dissemination of the virus as well as to elucidate the viral potential in animals and humans. The climatic changes affecting Earth’s ecosystems will most likely lead to the emergence of new viruses such as ZARV, which can challenge public health resources worldwide.

#### 3.2.2. Tick-Borne Encephalitis Virus Situation in Iran

Tick-borne encephalitis virus (TBEV) was first isolated in 1937 in Russia, and over 10,000 cases are reported each year [166]. In Europe and northern Asia, tick-borne encephalitis (TBE) is considered the predominant viral tick-borne zoonosis [167]. TBEV is a single stranded RNA virus with a genome on 11 kb with one open reading frame that belongs to genus *Flavivirus*. TBEV is transmitted mainly by hard ticks such as *Ixodes ricinus*, *Ixodes persulcatus*, *Haemaphysalis concinna*, and *Dermacentor reticulatus* with geographical location having an effect on which species disseminates the virus [168]. All three, *Ixodes*, *Haemephysalis*, and *Dermacentor* tick species have been identified in Iran [21,169]. The main reservoir of TBEV are small rodents [168], and Iran has a rich rodent diversity [32,170]. TBEV in Iran is not well documented and is often thought to be absent in Iran with just the vectors present. However, a recent study examining seroprevalence of the disease in the Mazadran province in northern Iran identified that 3.6% of the samples tested positive for TBEV, which provided evidence of infection in Iran [171] (Figure 3). Identification of TBEV in Iran is concerning, and tick populations in Iran should be monitored to prevent the further dissemination of this disease in the country, with studies examining human populations across Iran to elucidate distribution of the disease.

**Figure 3 microorganisms-11-03006-f003:**
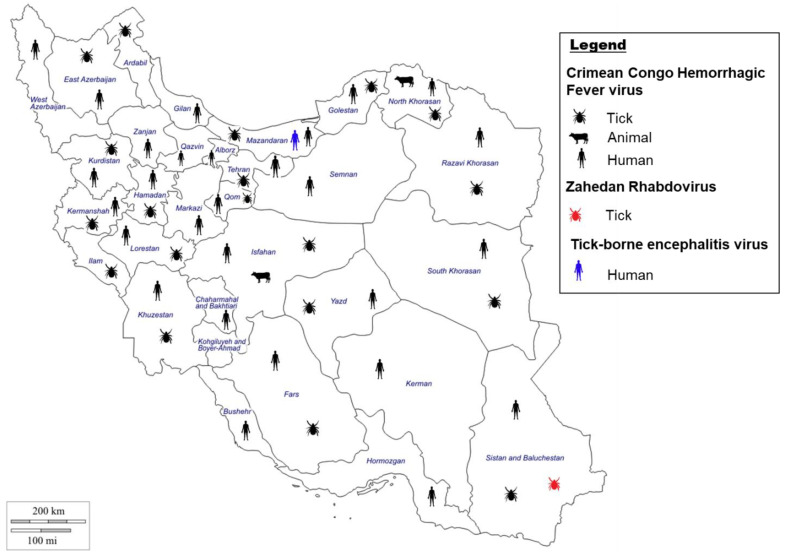
Evidence of Crimean–Congo Hemorrhagic Fever virus [23,132,137,139,172], Zahedan Rhabdovirus [165], and Tick-borne encephalitis virus [171] in Iran in either humans, animals, or ticks by indirect and direct evidences.

## 4. Sandfly-Borne Viruses in Iran

### 4.1. Sandfly-Borne Phleboviruses

Sandfly fever (SF) (also known as: phlebotomus fever, three-day fever or Pappataci fever) is caused by Phleboviruses, known as sandfly fever viruses (SFV), which are transmitted by phlebotomine sandflies [2,27,173]. SF is one of the most common viral diseases worldwide, causing a significant public health concern, especially as it impacts travellers and soldiers visiting endemic countries [28]. Sandfly-borne phleboviruses (SBPVs) belong to the genus *Phlebovirus* and are classified into four main serocomplexes, including Sandfly fever Naples viruses (SFNV), Sandfly fever Sicilian viruses (SFSV), Karimabad viruses (KARV), and Salehabad virus (SALV) [174,175]. Phleboviruses have a tri-segmented, negative-sense, single-stranded RNA genome [175]. The three segments, L, M, and S, encode the RNA-dependent RNA polymerase, the viral envelope glycoproteins, the viral nucleocapsid protein (N) and nonstructural protein (Ns).

SBPV infections are characterized as febrile illnesses with symptoms similar to influenza, with severe cases involving the disruption of the central nervous system resulting in signs such as trembling and unconsciousness [176]. SBPVs are extensively dispersed over Africa, the Mediterranean Basin, the Indian subcontinent, the Middle East, and the former USSR territories in the Far East [177]. Furthermore, as climate changes increase and vector ranges expand to new habitats, SBVs will most likely subsequently spread [176].

#### 4.1.1. Sandfly-Borne Viruses Situation in Iran

SBPV infections have been reported from the endemic regions of Turkey and Pakistan (sharing borders with Iran) and were responsible for several outbreaks in those countries [178,179]. In the 1970s, virus isolation showed four different SBVs present in Iran representing SFSV, SALV, KARV, and Tehran virus (TEHV belonging to SFNV serocomplex) (Figure 4). SFSV was isolated from an unidentified *Phlebotomus* spp. in 1975; SALV was isolated from an unidentified Phlebotomus spp. in 1959; KARV was first isolated from an unidentified pool of Phlebotomus spp. in 1959 and later from an unidentified pool of Phlebotomus spp. in 1975, where 99% of samples were *Phlebotomus papatasi*; and TEHV was isolated from *Phlebotomus perfiliewi* in 1976 [180]. For several decades, there was only indirect evidence of SBPV circulation in Iran. An earlier seroprevalence study conducted in the late 1980s in western Iran (Ilam and Kermanshah provinces) showed exposure to SFSV and SFNV in 60% and 46% of samples (from suspected patients), respectively [181]. More recently, SF has been documented in military personal along the western border of Iran (Ilam), whereby 18.4% of tested personal were positive for phlebovirus antibodies, providing indirect evidence of SFV circulation in Iran [173].

The increased investigation of SBVs in Iran resulted in the identification of a SFSV in both *Sergentomyia spp* and *Ph. papatasi*, coined the Dashli virus in Dashli Borun in northeastern Iran [27]. Similar efforts in molecular epidemiology of SBPVs in seven provinces of Iran revealed autochthonous transmission of SFSV by *Ph. papatasi*, TEHV by *Ph. sergenti*, and KARV by *Ph. papatasi* and *Ph. perfiliewi* in Western, north-western and central Iran [182]. These data demonstrate the need for further investigation into phleboviruses in Iran to potentially identify additional novel viruses distributed by sandflies.

#### 4.1.2. Sandfly-Borne Viruses Vectors in Iran

In the Old World, SBPVs were mainly transmitted by Phlebotomus sandflies, while in the New World, they are transmitted by Lutzomyia flies [183]. The subfamily *Phlebotominae* includes the genera *Phlebotomus* and *Sergentomyia*, which both bite humans, with Phlebotomus mainly feeding on mammals. Phlebotomine sandflies are widely distributed globally, including Asia, South America, and Australia, and with the effect of climate change, their distribution and colonization of new suitable habitats is predicted to increase [176]. The combination of several surveillance studies in Iran, indicates the presence of 48 sandfly species belonging to two genera of *Phlebotomus* (30 species) and *Sergentomyia* (18 species) [184,185] (Figure 4). *Ph. papatasi* was the most abundant species, present in all studied areas and most of the climatic zones, which poses a threat of transmission of SBPVs across the country [176]. Furthermore, the wide distribution of sandflies in Iran creates the opportunity for extended spread of current known SFVs [176], making the monitoring of sandflies of importance to mitigate the potential for spread of SF.

**Figure 4 microorganisms-11-03006-f004:**
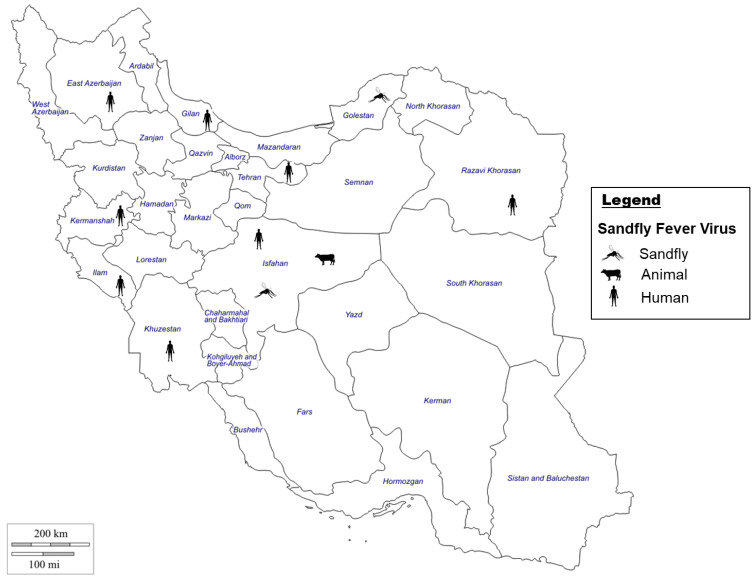
Evidence of Sandfly fever virus [2,27,28,173,180,186,187] in Iran in either humans, animals, or sandflies by indirect and direct evidences.

## 5. Rodent-Borne Viruses in Iran

### 5.1. Hantavirus

Hantaviruses are zoonotic *Bunyaviridae* viruses that are disseminated globally and are linked with the presence of their rodent hosts [188,189]. Hantavirus was first isolated in 1976 but has been recognized as a cause of disease since the 1950s [190]. They are generally grouped as Old and New World Hantaviruses, which have similar genetic organization [188,190,191] but cause 2 different types of diseases, hemorrhagic fever with renal syndrome (HFRS) and hantavirus cardiopulmonary syndrome (HCPS), respectively [189]. HFRS cases mainly occur in Asia and Europe with mortality rates of approximately 5–10%, while HCPS cases mainly occur in the United States and Canada and have a mortality rate of 40% [192,193,194]. Presently, over 40 hantavirus strains have been recognized but not all cause human disease, and only 22 are considered significant to public health [33].

Hantaviruses are a single-stranded RNA virus and the genome consists of three different segments consisting of L, M, and S, which encode the nucleoprotein (N), glycoprotein (Gn), and the viral envelope RNA dependent polymerase, and the L protein [189,195]. Hantavirus infection varies from mild asymptomatic physiologies to lethal outcomes [33,189]. Typically, signs and symptoms include fever, headaches, abdominal pain and nausea, among others. Hantavirus cases with kidney injuries or other signs of renal involvement suggest HFRS complications, and cases with respiratory failure or cardiac complications suggest HCPS etiologies [189]. Hantavirus is classically detected using serological tests examining antibodies but immunohistochemistry examining tissues and RT-PCR are also used to diagnose infections [196].

#### 5.1.1. Hantavirus Situation in Iran

Hantavirus is a rodent-borne virus (RBV) that is of concern worldwide, and Iran is home to a rich diversity of rodents, which are known reservoirs of Hantaviruses [32,170]. In Iran, Hantavirus was first identified in 2014 from serum samples collected in 2013 [197] (Figure 5). A study by Salehi-Vaziri et al., 2019, examined serum samples from patients in 25 different Iranian provinces, and 20.4% of the samples were positive using serological tests [33]. A subsequent study in 2021 examining street workers, thought to be a high-risk group, found 2 positive samples, though the individuals did not present with symptoms [198]. Based on current studies, Hantavirus appears to be circulating in Iran, and additional studies are needed to elucidate the role of Hantavirus in this country, toward enhancing mitigation and surveillance strategies.

**Figure 5 microorganisms-11-03006-f005:**
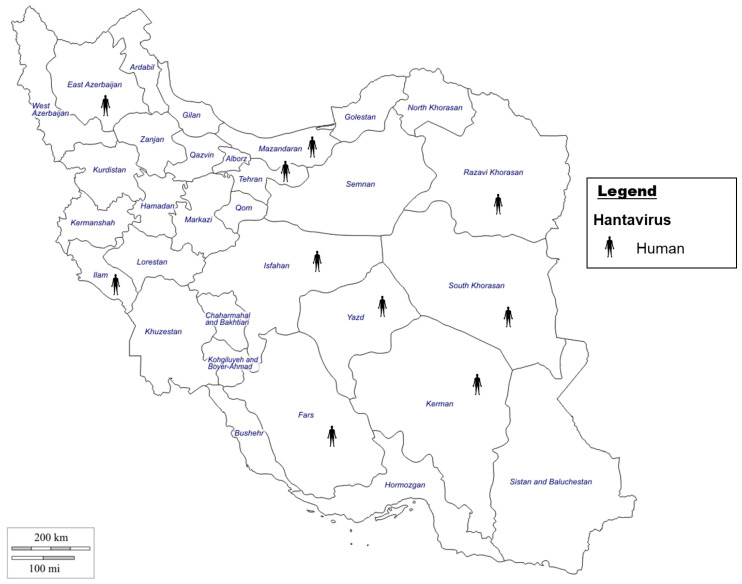
Evidence of Hantavirus [33,197,199] in Iran in humans by indirect and direct evidence.

#### 5.1.2. Hantavirus Vectors in Iran

Though many viruses are spread by insects, Hantaviruses are spread by rodents, either directly by bites or indirectly through inhaling germs in feces or ingestion of contaminated material [7,190]. While rodents are reservoirs of Hantaviruses, they do not exhibit any signs of disease but can spread the disease to humans [189,191]. In Iran, many diverse rodent populations exist including members of the *Muridae* and *Cricetidae* families, introduced as the reservoirs of the human pathogenic hantaviruses worldwide [7,32]. Furthermore, climate change and encroaching urbanization and agricultural expansion disrupt the many rodent habitats, causing the displacement, expansion, and overlap of rodent activity with humans, increasing transmission opportunities. Lastly, lax implementation of environmental and public health policies compounds the issue, increasing human contact with the disease reservoirs.

## 6. Conclusions

VBVDs are intriguing illnesses with vectors capable of disseminating viral, bacteria, or filarial diseases, creating a host of transmission considerations [13]. For example, the same vector is capable of transmitting more than one disease [13], leading to the potential of co-infections with two or more disease progressions in a patient [41], which clinically is not an obvious differential, as signs and symptoms are often attempted to be explained by a single causal agent. Alternatively, some diseases are capable of being transmitted by more than one vector. For example, RVFV, which is typically transmitted by mosquitoes, can also be spread by rodents, demonstrating both the versatility of VBVDs and the many challenges toward mitigating disease spread [7]. Alarmingly, these challenges are exacerbated by the growing rate of emerging VBDs, especially viral VBDs, ref. [1] due to a host of factors, both climatically and human instigated.

VBVDs are affected by various factors, such as temperature changes, which data suggests are correlated with increases in arthropod vectors and the subsequent VBVD spread [9], as well as non-climatic factors, such as urbanization [1]. For example, though a weak flier, colonization of the mosquito vector *Ae. albopictus* from neighboring countries into Iran is affected by climate changes involving mean temperature differences, creating spread into new urban and suburban environments [200]. This multifactorial consideration in VBVD transmission highlights the importance of a One Health approach in surveillance and response to vectors and their associated diseases in Iran.

Many mosquito species identified in Iran are known vectors for viral diseases, such as CHIKV, RVFV, SINV, and WNV [13,14]. A recent study in Iran determined that temperature has an effect on mosquito survival, especially over 30 °C [201]; hence, climatic warming can cause mosquitoes to expand their range and the diseases they spread.

Of the 46 tick species identified in Iran [21,23], *Hyalomma* ticks are considered the predominant vector of CCHF, with *H. marginatum* and *H. analoticum* among the most vital invasive species in Iran [11,23]. Tick populations increase as their ranges expand, and their suitable environments increase due to effects such as climate change [11,202]. Furthermore, the now higher temperatures in winter months extend the tick active periods and reduce their mortality rate [11,203]. Alternatively, a recent study in Iran demonstrated that *Rh. bursa* hard-bodied ticks were affected by a shift to colder temperatures, which altered viability and biodiversity. Such global climate changes are not limited to Iranian arthropods, but likely will affect tick populations worldwide [204], thus lessons learned in Iran will have translational value globally.

Phlebotomine sandflies are another important vector of viral disease in Iran and globally. Sandflies are small arthropods that live in environments in close contact with animals or humans, whether it be in the countryside or residential areas [28]. Sandflies as a vector are typically linked to leishmaniasis but they are also the vector for Sandfly fevers (SF) [2,27]. Phlebotomines are disseminated worldwide, including Asia, South America, and Australia. Furthermore, climate change likely will affect their distribution by increasing and introducing new suitable habitats [176]. In turn, sandfly spread will increase the spread of the diseases they carry and may lead to the identification of emerging viral diseases, as seen with the Dashli virus [27].

The ability of rodents to adapt to various environments make them successful pests/vectors across many climates and conditions. Rodent populations have been shown to recover quickly after floods, despite initially declining, potentially due to less predators or ability to find an area to hide from the flood waters, allowing for quick repopulation after waters recede [205]. Rodents quickly adapt to sudden environmental changes both as individuals and a species, fast repopulating after floods initially decimate their population, even having the ability to quickly introduce modifications to body size into their genetic distribution [206].

Iran is rich in diversity, from the geography and climate to the biodiversity of arthropod and animal vectors, as well as a diversity in human living, occupational, and travel activities that increase risk of disease exposure. The country is home to various VBDs and their vectors, and several countries sharing borders with Iran contain VBDs that have not yet been identified in Iran but remain of concern [2], necessitating enhanced surveillance. Furthermore, Iran is in proximity to Africa, home of several VBDs and vectors [22] that are a mere flight away, either through movement of migratory birds [24] or through international travel [207]. Additionally, climate change is projected to affect Iran, temperatures are expected to rise and precipitation is expected to decline [10], which can be anticipated to have an effect on VBVDs and vectors.

Lastly, projections of future rising temperatures and lower precipitation in Iran associated with climate change will have robust effects on VBVDs and their vectors [10], once again, highlighting the need for a vigilant One Health surveillance agenda that can adapt mitigation strategies as quickly as the VBVDs and their vectors adapt to novel environments. Not only do pathogen surveillance and mitigation strategies reduce the propensity for global spread, but the lessons learned from such a strategy in Iran address universal themes with global translational value.

## Data Availability

No new data were created or analyzed in this study. Data sharing is not applicable to this article.

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
