# Peer review of "Medically Significant Vector-Borne Viral Diseases in Iran"

_microorganisms, 2023, doi:10.3390/microorganisms11123006_

Round 1
Reviewer 1 Report
Comments and Suggestions for Authors
Vector-borne diseases, infectious and parasitic diseases of humans and animals, the pathogens of which are transmitted by blood-sucking arthropods (insects, ticks, flies, mosquitoes). Vector-borne infections include more than 200 nosological forms caused by viruses, bacteria, rickettsia, protozoa, and helminth larvae. Vector-borne diseases include such dangerous diseases as Dengue fever, Crimean hemorrhagic fever, West Nile fever, malaria, yellow fever, tularemia, typhus, tick-borne relapsing fever, tick-borne encephalitis, ixodid tick-borne borreliosis, etc. Every year, more than 700,000 people worldwide become ill with diseases such as malaria, dengue fever, schistosomiasis, human African trypanosomiasis, leishmaniasis, Chagas disease, yellow fever, Japanese encephalitis and onchocerciasis. In tropical and subtropical areas the burden of these diseases is particularly high, and the poorest people are disproportionately affected. Since 2014, large outbreaks of dengue, malaria, chikungunya, yellow fever and Zika virus disease have left many people injured or killed in several countries around the world, putting health systems under enormous strain. Other diseases, such as chikungunya, leishmaniasis and lymphatic filariasis, leave lasting consequences, doom people to suffering throughout their lives, and lead to disability. The distribution of vector-borne diseases is determined by a complex of demographic, environmental and social factors.
The authors describe in detail vector-borne diseases in Iran. The names of viruses and vectors are given and maps of Iran are detailed. I recommend that each species (vectors and viruses) that is written for the first time write the name in full. And then you can give them in an abbreviated version. For example.... Aedes aegypti, Aedes albopictus and than......Ae. aegypti, Ae. albopictus.
The manuscript may be accepted for publication.
Author Response
Thank you for your comment! The names of vectors and viruses has been corrected in the revised version of the manuscript.
Reviewer 2 Report
Comments and Suggestions for Authors
The review manuscript focuses on the vector-borne viral diseases (VBVDs) in Iran in which the authors examined the epidemiology of numerous VBVDs occurring in Iran and the vectors responsible for their transmission including mosquitoes, ticks, sandflies, and rodents. It is a well written review paper with very helpful data about the vector borne viral diseases in Iran. A few areas in the manuscript need grammatical improvements - so please review the grammar in more detail. The map figures need improvement by shading areas where the VBVDs occur as opposed to placing a symbol in the province. Overall, the review paper reads well.
Comments on the Quality of English LanguageQuality of English acceptable
Author Response
A few areas in the manuscript need grammatical improvements - so please review the grammar in more detail.
Response: Thank you for your comment! The author that is a native English speaker, has revised the article for any possible gramatical errors.
The map figures need improvement by shading areas where the VBVDs occur as opposed to placing a symbol in the province.
Response: Thank you fo your comment! We first used shaded areas for VBDs in our original version of the figures. However, this did not work well because we felt that readers would find it confusing when many VBDs were included in one shaded area. We then decided to use symbols and create the the current version of the map and we feel strongly that this is the best representation of the data.
Reviewer 3 Report
Comments and Suggestions for Authors
The manuscript is about "Vector-borne viral diseases of public health importance in Iran".
Abstract: It is general. It does not show the depth of the review. Kindly add some results and analysis from review in the abstract, how many mosquitoes, ticks, sandflies, and rodents diseases authors have documented?
Introduction: Authors kindly write the aim of the review.
Kindly provide a table on epidemiology of said vectors, viruses and vector-borne diseases in different areas of Iran in different time periods.
Overall manuscript is written well. It provides comprehensive information on viral diseases in Iran. However, authors did not provide any information how to manage these viral diseases, control vectors, fight with climate change to curb spread of these vectors, pathogens and diseases.
Is there any drugs or vaccines available to manage these diseases?
Authors may provide information on transmission cycle of diseases.
Which key diseases are widely spread in the Iran?
Conclusion is too long.
Comments on the Quality of English Language
Minor English language errors.
Kindly proofread carefully.
Author Response
Abstract: It is general. It does not show the depth of the review. Kindly add some results and analysis from review in the abstract, how many mosquitoes, ticks, sandflies, and rodents diseases authors have documented?
Response: Thank you for your comment! The diseases we have documented in the review article are added to the new version of abstract, we are limited by the word limit and have expanded as much as possible.
Introduction: Authors kindly write the aim of the review.
Response: Thank you for your comment! The aim of the review is expanded in the revised version of the introduction.
Kindly provide a table on epidemiology of said vectors, viruses and vector-borne diseases in different areas of Iran in different time periods.
Response: This is an excellent comment! However, in order to provide a spatiotemporal table for VBDs in Iran, such data must first be generated by a separate original study. We just completed a review study using data on VBDs in Iran that was already available.
authors did not provide any information how to manage these viral diseases, control vectors, fight with climate change to curb spread of these vectors, pathogens and diseases.
Response: Thank you for your comment! Regarding your concern, we address a few possible ways to manage viral diseases more effectively in the final paragraph of the discussion, including
- vigilant One Health surveillance agenda
- pathogen surveillance and mitigation strategies
- lessons learned from such a strategy
As researchers, we think we are not in the position to offer management advice. We can just share our knowledge with public health authorities, and they have the responsibility to create policies to combat climate change and emerging diseases, we may share our knowledge with them.
Is there any drugs or vaccines available to manage these diseases?
Response: Thank you for your question! For the majority of emerging zoonotic diseases there is yet no effective therapeutic approaches or vaccine. This highlight the importance of early detection and characterization of emerging diseases.
Authors may provide information on transmission cycle of diseases.
Response: Thank you for your comment! The transmission cycle of diseases is provided in the manuscript. For example, in line 160 for CHIKV, line 273 for SINDV, line 337 forl WNV, in line 405 for CCHFV
Which key diseases are widely spread in the Iran?
Response: Thank you for your question! CCHFV is widely spread in Iran
Conclusion is too long.
Response: Thank you for your comment! Since other reviewers are OK with discussion length, to have a consensus amongst all the opinions from each reviewer, we would leave the decision of deleting some parts of the discussion to the Editor.
Round 2
Reviewer 3 Report
Comments and Suggestions for Authors
No comments
Comments on the Quality of English LanguageKindly proofread carefully